# Improving the Minimum Free Energy Principle to the Maximum Information Efficiency Principle

**DOI:** 10.3390/e27070684

**Published:** 2025-06-26

**Authors:** Chenguang Lu

**Affiliations:** 1Intelligence Engineering and Mathematics Institute, Liaoning Technical University, Fuxin 123000, China; survival99@gmail.com; 2School of Computer Engineering and Applied Mathematics, Changsha University, Changsha 410022, China

**Keywords:** variational Bayes, free energy principle, Shannon mutual information, semantic mutual information, information rate-fidelity, EM algorithm, active inference, entropy, free energy, Boltzmann distribution

## Abstract

Friston proposed the Minimum Free Energy Principle (FEP) based on the Variational Bayesian (VB) method. This principle emphasizes that the brain and behavior coordinate with the environment, promoting self-organization. However, it has a theoretical flaw, a possibility of being misunderstood, and a limitation (only likelihood functions are used as constraints). This paper first introduces the semantic information G theory and the *R*(*G*) function (where *R* is the minimum mutual information for the given semantic mutual information *G*). The G theory is based on the P-T probability framework and, therefore, allows for the use of truth, membership, similarity, and distortion functions (related to semantics) as constraints. Based on the study of the *R*(*G*) function and logical Bayesian Inference, this paper proposes the Semantic Variational Bayesian (SVB) and the Maximum Information Efficiency (MIE) principle. Theoretic analysis and computing experiments prove that *R* − *G* = *F* − *H*(*X*|*Y*) (where *F* denotes VFE, and *H*(*X*|*Y*) is Shannon conditional entropy) instead of *F* continues to decrease when optimizing latent variables; SVB is a reliable and straightforward approach for latent variables and active inference. This paper also explains the relationship between information, entropy, free energy, and VFE in local non-equilibrium and equilibrium systems, concluding that Shannon information, semantic information, and VFE are analogous to the increment of free energy, the increment of exergy, and physical conditional entropy. The MIE principle builds upon the fundamental ideas of the FEP, making them easier to understand and apply. It needs to combine deep learning methods for wider applications.

## 1. Introduction

In 1993, Hinton et al. [1,2] employed the minimum free energy as an optimization criterion to improve neural network learning, resulting in significant breakthroughs. This method was later developed into the Variational Bayesian (VB) approach [3,4], which has since been widely and successfully applied in machine learning (including reinforcement learning) [5]. Friston [6,7] extended the application of VB to neuroscience, evolving the Minimum Free Energy (MFE) criterion into the Minimum Free Energy Principle (FEP), aiming for it to become a unified scientific theory of brain function and biological behavior. This theory integrates concepts such as predictive coding, perceptual prediction, active inference, information, and stochastic dynamical systems [7,8,9]. Unlike the passive perspective in the book *Entropy: A New World View* [10], Friston’s theory inherits the optimistic entropy-based worldview from the existing Evolutionary Systems Theory (EST) [11,12] and emphasizes the ability of bio-organisms to predict, adapt to, and influence their environments, which promotes self-organization and order.

Friston’s theory has garnered widespread attention and deep reflection [13]. Many applications have emerged [13,14]. Information and free energy have been combined to explain self-organization and order [15,16]. However, some criticisms have also been raised. Some argue that FEP is valuable as a tool for machine learning; however, its validity as a universal principle remains debatable. What is more, some think that free energy means negative entropy and is essential for life; thus, minimizing free energy would imply death [17]. These criticisms are mainly due to the confusion between Variational Free Energy (VFE) and physical free energy.

Another controversy is whether the FEP is another form of the Maximum Entropy (ME) principle. As early as 1990, Silverstein and Pimbley [18] used the “minimum free energy method” in the title of their article. Their objective function is defined as a linear combination of the mean-square error energy expression and the signal entropy expression. In the paper entitled *Two Kinds of Free Energy and the Bayesian Revolution*, Gottwald and Braun [19] reviewed many similar studies and argued that Friston’s free energy is just one of the two kinds. The other kind, including free energy in [18], is expressed as various objective functions in the Maximum Entropy (ME) method. These objective functions either equal the reward function plus the entropy function (which needs to be maximized) or the average loss minus the entropy function (which needs to be minimized). The authors believe both kinds of free energy methods are very significant.

In my opinion, Friston’s FEP is different from the ME principle [20,21] for the following reasons:Bio-organisms have purposes and can predict environments and control outcomes. The FEP can express subjective and objective coincidence. The subjective and objective approaches are bidirectional and dynamic.Loss functions are expressed in terms of information or entropy measures, transforming the active inference issue into the inverse problem of sample learning.There exist multi-task coordination and tradeoffs requiring the inference of latent variables.

It appears that the FEP can better explain the subjective initiative of bio-organisms than the ME principle. However, from the author’s perspective, the FEP is still imperfect, as well as VB. It needs improvements for three reasons:Its convergence theory is flawed;It is prone to cause misunderstanding because of the use of “free energy”;It has a limitation in that it only uses likelihood functions as constraints and utilizes the KL divergence between the subjective and objective probability distributions to assess how well they match when the bio-organism predicts and adapts to (including intervenes in) the environment.

In other words, the FEP has three defects: a theoretical flaw, a possibility of being misunderstood, and a limitation.

These reasons come from the author’s research on semantic information theory. Thirty years ago, the author generalized Shannon information theory to obtain a semantic information theory [22,23,24]. Shannon’s and semantic mutual information (MI) formulas are as follows:*I*(*X*; *Y*) = *H*(*X*) − *H*(*X*|*Y*),(1)*I*(*X*; *Y_θ_*) = *H*(*X*) − *H*(*X*|*Y_θ_*),(2)
where *X* and *Y* are two random variables denoting an instance and a label (or an observed datum and a latent variable), respectively. *H*(*X*) is the source entropy, *H*(*X*|*Y*) is the Shannon conditional entropy, and *H*(*X*|*Y_θ_*) is the semantic conditional entropy or conditional cross-entropy (see Section 3.2 for details). Roughly speaking, *H*(*X*|*Y_θ_*) equals VFE (see Section 6.1 for details). Hence, minimizing VFE is equivalent to maximizing semantic MI.

Later, the author referred to the generalized theory as the Semantic Information G Theory [25,26,27] (or simply G Theory, where “G” stands for “generalization”). The author recognized early that *H*(*X*|*Y_θ_*) does not necessarily decrease monotonically when the Shannon channel matches the semantic channel. The author also studied mixture models with the Expectation–Maximization (EM) algorithm [25]. Experimental observations indicated that *H*(*X*|*Y_θ_*) and VFE do not continually decrease as the mixture model converges. Although VB ensures the mixture model’s convergence [2,3], its theoretical justification is flawed. This is the first reason for the improvement.

The second reason exists because, in physics, free energy is the energy available to perform work and is usually maximized [28]. If we actively minimize free energy, are we not simply following the trend of increasing entropy [17]? Recently, Friston et al. analyzed VFE in biological adaptive systems and observed a nearly monotonic decrease in VFE as the system relaxes to a steady state [29,30,31]. This is true, but it is also true that at the same time, physical free energy increases. Friston et al. also concluded [32] that usually, low VFE correlates with high thermodynamic free energy. If so, it can be better understood to say that what VFE represents is semantic conditional entropy or conditional cross-entropy, which are inversely correlated to physical free energy.

As early as 1993, the author [23] analyzed information between temperature and a molecule’s energy in local non-equilibrium and equilibrium systems. The conclusions suggest that Shannon MI is analogous to the increment of free energy in a local non-equilibrium system, while semantic MI is comparable to the increment of free energy in a local equilibrium system. To better explain the second reason, this paper will clarify what VFE, or *H(X*|*Y_θ_*), truly represents in a thermodynamic system and how information, entropy, free energy, and semantic conditional entropy (or VFE) mutually relate within such a system (see Section 5).

The third reason for the improvement is that, in addition to likelihood functions, we can also employ truth, membership, similarity, and distortion functions as constraints and use them to express semantic information, reflecting both subjective and objective alignments, thereby expanding the application scope. To measure semantic information, G theory adopts the P-T probability framework [26], including the statistical probability represented by *P* and the logical probability by *T*. A conditional logical probability is a truth or membership function, which means the extension or denotation of a concept and therefore reflects a label’s semantics or a bio-organism’s purpose.

Before improving FEP, we must improve VB. Also in 1993 [23], the author extended Shannon’s rate distortion function *R*(*D*) [33] to obtain the information rate-fidelity function *R*(*G*) [23,25], where *R* represents the minimum Shannon MI, *I*(*X*; *Y*), for given semantic MI, *G* = *I*(*X*; *Y_θ_*), (representing fidelity). The Semantic Variational Bayesian (SVB) [34], an improved version of VB, operates the Minimum Information Difference (MID) iteration for optimizing the Shannon channel *P*(*y|x*) for the *R(G*) function. SVB can resolve the issues VB addresses, but does not always minimize VFE. Instead, it minimizes *R* − *G* = *F* − *H*(*X*|*Y*). Minimizing *R* − *G* is equivalent to maximizing information efficiency: *G*/*R*. Thus, the author proposes the Maximum Information Efficiency (MIE) principle as the improved version of the FEP.

The applications of G theory [27] in machine learning include multi-label learning, maximum MI classification for unseen instances, mixture models [25], Bayesian confirmation [26], solving latent variables [34], and semantic compression [35]. The success of these applications strengthens the validity of G theory.

The motivation of this paper is to clarify some defects in VB and the FEP. The aim is to provide an improved version of the FEP for better explaining how bio-organisms efficiently utilize information and free energy to predict and influence the environment, promoting self-organization and order.

The contributions of this paper:Mathematically clarifying the theoretical and practical inconsistencies in VB and the FEP.Explaining the relationships between Shannon MI, semantic MI, VFE, physical entropy, and free energy from the perspectives of G theory and statistical physics.Providing experimental evidence by mixture model examples (including one used by Neal and Hinton [2]) to demonstrate that VFE may increase during the convergence of mixture models and explain why this occurs.

The iteration method for minimum information difference in SVB is inspired by the iterative approach used by Shannon et al. in solving the Shannon channel *P*(*y*|*x*) for the rate-distortion function [33,36,37].

All the abbreviations with original texts are listed in the back matter. Information about the Python 3.6 source code for producing most figures in this paper can be found in Appendix A.

## 2. Two Typical Tasks of Machine Learning

### 2.1. Sheep Clustering: Mixture Models

To explain the tasks to be completed by VB. Let us take sheep clustering and sheep herding as examples. This section describes the sheep clustering issue (see Figure 1).

Suppose several sheep flocks are distributed on a grassland with fuzzy boundaries. We can observe that the density distribution (the proportion of the number of sheep per unit area) is *P*(*x*). We also know that there are *n* flocks of sheep, and their distributions exhibit a certain regularity, such as the Gaussian distribution. We can establish a mixture model: *P_θ_*(*x*) = *P*(*y*_1_)*P*(*x*|*θ*_1_) + *P*(*y*_2_)*P*(*x*|*θ*_2_) +… where *P*(*x*|*θ_j_*) = *P*(*x*|*y_j_, θ*). Then, we use the maximum likelihood criterion or the minimum cross-entropy criterion to optimize the mixture ratios *P(y*_1_), *P(y*_2_), … and the model parameters.

The Expectation–Maximization (EM) algorithm [38,39] is typically used to solve mixture models (i.e., to optimize their parameters). This algorithm is very clever. It can automatically adjust the different components, namely, the likelihood function for each cluster (circled in Figure 1), so that each component covers a group of instances (a flock) and can provide the appropriate ratios *P*(*y_j_*), *j* = 1, 2, 3, 4, of the mixture model’s components, also known as the probability distribution of the latent variable. We sometimes refer to *P*(*y*) as the latent variable.

The EM algorithm is not ideal in two aspects: (1) there have been problems with its convergence proof [25], which has led to blind improvements; and (2) *P*(*y*) sometimes converges very slowly, and it is challenging to solve *P*(*y*) when the likelihood functions remains unchanged. Researchers use VB not only to improve the EM algorithm [2,3] but also to solve latent variables [5].

The mixture model belongs to unsupervised learning in machine learning and is very representative. Similar methods can be found in Restricted Boltzmann Machines and deep learning pre-training tasks.

### 2.2. Driving Sheep to Pastures: Constrained Control and Active Inference

Driving sheep to pastures is a constraint control issue involving random events and also an active inference issue related to reinforcement learning. In this case, Shannon MI reflects the control complexity. We need to maximize the purposefulness (or utility) and minimize the Shannon MI, the control cost.

The circles in the figure represent the control targets; the points in Figure 2a reflect the initial flock density distribution *P*(*x*). There are usually two types of control objectives or constraints:The objectives are expressed by the probability distributions *P*(*x*|*θ_j_*) (*j* = 1, 2, 3, 4). Given *P*(*x*) and *P*(*x*|*θ_j_*), we solve the Shannon channel *P*(*y*|*x*) and the herd ratio *P*(*y*). It is required that *P*(*x*|*y_j_*) is close to *P*(*x*|*θ_j_*) and that *P(y)* minimizes the control cost.The objectives are expressed by the fuzzy ranges. *P*(*x*|*θ_j_*) (*j* = 1, 2, …) can be obtained from *P*(*x*) and the fuzzy ranges, and the others are the same.

Circles in Figure 2 represent constraint ranges. The difference between clustering in mind (see Figure 1) and herding in reality (see Figure 2): for clustering, *P*(*x*) is fixed, and the herd ratios are objective; for herding, *P*(*x*) is transferred to the target areas, and the herd ratios are adjusted according to the minimum control cost criterion. For example, when clustering, the proportions of the two groups of sheep in the middle are larger; when herding, the proportions of the two groups on the sides are larger because it is easier to drive sheep there. In addition, the centers of four groups of sheep in Figure 2b deviate from the target centers, which is also for the sake of control cost. We can also increase the constraint strength to drive the sheep to the ideal position. However, the control cost must also be considered. Both sheep clustering and herding require solving latent variables.

## 3. The Semantic Information G Theory and the Maximum Information Efficiency Principle

### 3.1. The P-T Probability Framework

The P-T probability framework is denoted by a five-tuple (***U***, ***V***, ***B***, *P*, *T*), where

***U*** = {*x*_1_, *x*_2_, …} is a set of instances, *X*∈***U*** is a random variable;***V*** = {*y*_1_, *y*_2_, …} is a set of labels or hypotheses, *Y*∈***V*** is a random variable;***B*** = {*θ*_1_, *θ*_2_, …} is the set of subsets of ***U***; every subset *θ_j_* has a label *y_j_*∈***V***;*P* is the probability of an element in ***U*** or ***V***, i.e., the statistical probability as defined by Mises [40] with “=“, such as *P*(*x_i_*) = *P*(*X = x_i_*);*T* is the probability of a subset of ***U*** or an element in ***B***, i.e., the logical probability as defined by Kolmogorov [41] with “∈”, such as *T*(*y_j_*) *= P*(*X*∈*θ_j_*).

In addition, we assume *θ_j_* is a fuzzy set [42] and also a model parameter.

The truth value of *y_j_* for given *x* is the membership grade of *x* in *θ_j_*, which is also the conditional logic probability of *y_j_*, namely:*T*(*y_j_*|*x*) ≡ *T*(*θ_j_*|*x*) ≡ *m_θ__j_* (*x*).(3)

Why do we need the P-T probability framework? The reasons include

The P-T probability framework enables us to use truth, membership, similarity, and distortion functions as constraints for solving latent variables in wider applications.The truth function can represent a label’s semantics or a bio-organism’s purpose and can be used to make semantic probability predictions [27];

According to Davidson’s truth-conditional semantics [43], *T*(*y_j_*|*x*) reflects the semantics of *y_j_*. The logical and statistical probabilities of a label are often unequal. For example, the logical probability of a tautology is 1, while its statistical probability is close to 0. We have *P*(*y*_1_) + *P*(*y*_2_) + … + *P*(*y_n_*) = 1, but it is possible that *T*(*y*_1_) + *T*(*y*_2_) + … + *T*(*y_n_*) > 1.

According to the above definition, we have the following:(4)$$T(y_{j})\equiv T(\theta_{j})\equiv P(X\in \theta_{j})=\sum_{i}P(x_{i})T(\theta_{j}|x_{i}).$$

As we will see later, *T*(*y_j_*) is both the statistical physics partition function and the machine learning regularization term. We can put *T*(*θ_j_*|*x*) and *P*(*x*) into Bayes’ formula to obtain the semantic probability prediction formula [25]:(5)$$P(x|\theta_{j})=\frac{T(\theta_{j}|x)P(x)}{T(\theta_{j})},\;T(\theta_{j})=\sum_{i}T(\theta_{j}|x_{i})P(x_{i}).$$

*P*(*x*|*θ_j_*) is the likelihood function *P*(*x*|*y_j_*, *θ*) in the popular method. We refer to the above formula as the semantic Bayes’ formula.

Just as a set of transition probability functions *P*(*y_j_*|*x*) (*j* = 1, 2, …) constitutes a Shannon channel, a set of truth functions *T*(*θ_j_*|*x*) (*j* = 1, 2, …) constitutes a semantic channel. The two kinds of optimization in VB and SVB can be explained as the mutual matching between the two channels.

The truth function and the distortion function can be converted to each other by defining [35] the following:*T*(*y_j_*|*x*) ≡ exp[−*d*(*x*, *y_j_*)], *d*(*x*, *y_j_*) ≡ −log*T*(*y_j_*|*x*).(6)

### 3.2. The Semantic Information Measure 

Shannon MI can be expressed as(7)$$I(X;Y)=\sum_{j}\sum_{i}P(x_{i})P(x_{i}|y_{j})\log\frac{P(x_{i}|y_{j})}{P(x_{i})}=H(X)-H(X|Y).$$

We replace *P*(*x_i_*|*y_j_*) on the right side of the log with the likelihood function *P*(*x_i_*|*θ_j_*), leaving the left *P*(*x*|*y_j_*) unchanged. Hence, we obtain semantic MI:(8)$$\begin{array}{l} I(X;Y_{\theta })=\sum_{j}\sum_{i}P(x_{i})P(x_{i}|y_{j})\log\frac{P(x_{i}|\theta_{j})}{P(x_{i})}\\ =\sum_{j}\sum_{i}P(x_{i})P(x_{i}|y_{j})\log\frac{T(\theta_{j}|x_{i})}{T(\theta_{j})}\\ =H(X)-H(X|Y_{\theta })=H(Y_{\theta })-H(Y_{\theta }|X)=H(Y_{\theta })-\bar{d}, \end{array}$$
where *H*(*Y_θ_*|*X*) is fuzzy entropy, which equals the average distortion $\bar{d}$ because, according to Equation (6), there is(9)$$H(Y_{\theta }|X)=-\sum_{j}\sum_{i}P(x_{i},y_{j})\log T(\theta_{j}|x_{i})=\bar{d}.$$

*H*(*X*|*Y_θ_*) is the semantic posterior entropy of *x*:(10)$$H(X|Y_{\theta })=-\sum_{j}\sum_{i}P(x_{i},y_{j})\log P(x_{i}|\theta_{j}).$$

Roughly speaking, *H*(*X*|*Y_θ_*) is VFE (*F*) in VB and the MFE principle. *H*(*Y_θ_*) is the semantic entropy:(11)$$H(Y_{\theta })=-\sum_{i}P(y_{j})\log T(\theta_{j}).$$

Semantic MI is less than or equal to the Shannon MI and reflects the average code length saved due to the semantic prediction. The maximum semantic MI criterion is equivalent to the maximum likelihood criterion and shares similarities with the Regularized Least Squares (RLSs) criterion. Semantic entropy *H_θ_*(*Y*) is the regularization term. Fuzzy entropy *H*(*Y_θ_*|*X*) is a more general average distortion than the average square error.

Suppose the truth function becomes a similarity function. In that case, the semantic MI becomes the estimated MI. The estimated MI has been utilized by deep learning researchers for Mutual Information Neural Estimation (MINE) [44] and Information Noise Contrast Estimation (InfoNCE) [45].

A simple example of the estimated information is the information conveyed by the GPS pointer. The truth function is also the similarity function, which can be represented by a Gaussian function. Suppose the real position is *x_i_*, and the pointed position is *y_j_* = ${\hat{x}}_{j}$, the truth function is *T*(*θ_j_*|*x_i_*) = exp[−(*x_j_* − *x_i_*)^2^/(2*σ*^2^)], the (amount of) estimated information is(12)$$I(x_{i};\theta_{j})=\log\frac{T(\theta_{j}|x_{i})}{T(\theta_{j})}=\log[1/T(\theta_{j})]-{\left(x_{j}-x_{i}\right)}^{2}/(2\sigma^{2}).$$

Semantic information conveyed by *y_j_* about *x_i_* is illustrated in Figure 3.

If *T*(*θ_j_*|*x_i_*) is always 1, the above semantic information measure (the G measure) becomes Carnap and Bar-Hillel’s semantic information measure [25]. The G measure reflects Popper’s idea of the philosophy of science [26].

### 3.3. From the R(D) Function to the R(G) Function

Shannon [33] defines that given a source *P*(*x*), a distortion function *d*(*y*|*x*) (when using *y* to represent *x*), and the upper limit *D* of the average distortion $\bar{d}$, we change the channel *P*(*y*|*x*) to find the minimum MI, *R*(*D*). *R*(*D*) is the information rate-distortion function. The minimum distortion criterion is like the criterion that “no fault is virtue”; however, what we need is a criterion that “merit outweighing fault is virtue.”

Now, we replace *d*(*y_j_*|*x_i_*) with *I*(*x_i_*; *θ_j_*) = log[*T*(*θ_j_*|*x_i_*)/*T*(*θ_j_*)], replace $\bar{d}$ with *I*(*X*; *Y_θ_*), and replace *D* with the lower limit, *G,* of the semantic MI to find the minimum Shannon MI, *R*(*G*). *R*(*G*) is the information rate-fidelity function. Solving the Shannon channel *P*(*y*|*x*) for *R*(*D*) or *R*(*G*) is similar to active inference, as both require finding the latent variable *P*(*y*).

Because *G* reflects the average code length saved due to semantic prediction, using *G* as the constraint is more consistent in shortening the code length, and *G*/*R* can better represent information efficiency.

The *R*(*G*) function is defined as(13)$$R(G)=\underset{P(y|x):I(X;\theta )\geq G}{\min}I(X;Y).$$

We use the Lagrange multiplier method to find the minimum MI. The constraint conditions include *I*(*X*; *Y_θ_*) ≥ *G* and(14)$$\sum_{j}P(y_{j}|x_{i})=1,\;i=1,\;2,\;\ldots ;\sum_{j}P(y_{j})=1.$$

The Lagrangian function is as follows:(15)$$L(P(y|x),P(y))=I(X;Y)-sI(X;Y_{\theta })-\mu_{i}\sum_{j}P(y_{j}|x_{i})-\alpha \sum_{j}P(y_{j})$$

Using *P*(*y*|*x*) as a variation, we let $\partial L/\partial P(y_{j}|x_{i})=0$. Then, we obtain(16)$$P*(y_{j}|x_{i})=P(y_{j})m_{ij}^{s}/\lambda_{i},\;\lambda_{i}=\sum_{j}P(y_{j})m_{ij}^{s},\;i=1,\;2,\ldots ;\;j=1,\;2,\ldots$$
where *m_ij_* = *P*(*x*_i_|*θ_j_*)/*P*(*x*_i_) = *T*(*θ_j_*|*x_i_*)/*T*(*θ_j_*). Using *P*(*y*) as a variation, we let $\partial L/\partial P(y_{j})=0$. Then, we obtain(17)$$P*(y_{j})=\sum_{i}P(x_{i})P(y_{j}|x_{i}),$$
where *P**(*y_j_*) means the next *P*(*y_j_*). Because *P**(*y*|*x*) and *P**(*y*) are interdependent, we can first assume a *P*(*y*) and then repeat the above two formulas to obtain convergent *P**(*y*) and *P**(*y*|*x*) (see [37] (P. 326)). We refer to this method as the Minimum Information Difference (MID) iteration. Someone may ask: Why do we obtain Equation (17) by variational methods instead of directly using Equation (17)? The answer is that if we use Equation (17) directly, we still need to prove that *P**(*y*) reduces *R* − *G.*

The parameter solution of the *R*(*G*) function (as illustrated in Figure 4) is as follows:(18)$$\begin{array}{l} G(s)=\sum_{i}\sum_{j}P(x_{i})P(y_{j}|x_{i})I_{ij}=\sum_{i}\sum_{j}I_{ij}P(x_{i})P(y_{j}){m_{ij}}^{s}/Z_{i},\\ R(s)=sG(s)-\sum_{i}P(x_{i})\log Z_{i},{\;Z}_{i}=\sum_{k}P(y_{k}){m_{ij}}^{s}. \end{array}$$

Any *R*(*G*) function is bowl-shaped (possibly not symmetrical) [24], with the second derivative greater than 0. The *s* = d*R*/d*G* is positive on the right. When *s* = 1, *G* equals *R*, meaning the semantic channel matches the Shannon channel. *G*/*R* represents information efficiency, with a maximum value of 1. *G* has a maximum value, *G*^+^, and a minimum value, *G*^−^, for a given *R*. *G*^−^ means how small the semantic information the receiver receives can be when the sender intentionally lies.

### 3.4. Semantic Variational Bayes and the Maximum Information Efficiency Principle

Using *P*(*x*|*θ_j_*) as the variation and letting $\partial L/\partial P(x_{i}|\theta_{j})=0$, we get *P*(*x*|*θ_j_*) = *P*(*x*|*y_j_*) or *T*(*θ_j_*|*x*) ∝ *P*(*y*_j_|*x*), which is the result of Logical Bayesian Inference (LBI) (see Appendix B). Therefore, the MID iteration plus LBI equals SVB. When using SVB to infer the latent variables, the constraint functions can be likelihood functions, truth (or membership) functions, similarity functions, and distortion functions (or loss functions). When the constraint is a set of likelihood functions, the MID iteration formulas are as follows:(19)$$\begin{array}{l} P*(y_{j}|x_{i})=P(y_{j}){\left[P(x_{i}|\theta_{j})\right]}^{s}/Z_{i},{\;Z}_{i}=\sum_{j}P(y_{j}){\left[P(x_{i}|\theta_{j})\right]}^{s},\\ \;P*(y_{j})=\sum_{i}P(x_{i})P(y_{j}|x_{i}). \end{array}$$

This formula, with *s* = 1, is the same as the formula used in the E-step of the EM algorithm. However, the EM algorithm only uses likelihood functions.

When the constraint is a set of truth or similarity functions, Equation (19) becomes the following:(20)$$\begin{array}{l} P*(y|x_{i})=P(y){\left[\frac{T(\theta_{j}|x_{i})}{T(\theta_{j})}\right]}^{s}/Z_{i},\;Z_{i}=\sum_{k}P(y_{k}){\left[\frac{T(\theta_{k}|x_{i})}{T(\theta_{k})}\right]}^{s},\\ \;P*(y_{j})=\sum_{i}P(x_{i})P(y_{j}|x_{i}). \end{array}$$

The hyperparameter *s* allows us to strengthen the constraint, thereby reducing the ambiguity of classification boundaries.

Using the MID criterion means using the MIE criterion. Applying this criterion to various fields amounts to using the Maximum Information Efficiency (MIE) principle. Note that the MIE principle not only maximizes *G*/*R*, it also maximizes *G*/*R* under the condition that *G* meets the requirement. This requires us to balance between maximizing *G* and maximizing *G*/*R*. As a constraint condition, *G* is not only about limiting the amount of semantic information but also about specifying what kind of semantic information is needed. It is related to human purposes and needs. Therefore, the MIE principle is associated with information values.

## 4. The MIE Principle for Mixture Models and Constrained Control

### 4.1. New Explanation of the EM Algorithm for Mixture Models (About Sheep Clustering)

The EM algorithm [41,42] is typically used for mixture models, a type of unsupervised learning (clustering) method. We know that *P*(*x*) = ∑*_j_ P*(*y*_j_) *P*(*x*|*y_j_)*. Given a sample distribution *P*(*x*), we use *P_θ_*(*x*) = ∑*_j_ P*(*x*|*θ_j_*)*P*(*y*_j_) to approximate *P*(*x*) so that the relative entropy or KL divergence *KL*(*P‖P_θ_*) is close to 0.

The EM algorithm first presets *P*(*x*|*θ_j_*) and *P*(*y*_j_). The E-step obtains the following:(21)$$P(y_{j}|x)=P(y_{j})P(x|\theta_{j})/P_{\theta }(x),\;P_{\theta }(x)=\sum_{k}P(y_{k})P(x|\theta_{k}).$$
This equation, with Equation (22), is similar to Equation (19) for the *R*(*G*) function.

In the M-step, the log-likelihood of the complete data (usually represented by *Q*) is maximized. The M-step can be divided into two steps: the M1-step for(22)$$P^{+1}(y_{j})=\sum_{i}P(x_{i})P(y_{j}|x_{i}),$$
and the M2-step for(23)$$P(x|{\theta_{j}}^{+1})=P(x)P(y_{j}|x)/P^{+1}(y_{j}).$$

For Gaussian mixture models, we can use the expectation and standard deviation of *P*(*x*)*P*(*y_j_*|*x*)/*P^+1^*(*y_j_*) as those of *P*(*x*|*θ_j_*^+1^). *P*^+1^(*y*) is the above *P**(*y*).

From the perspective of G theory, the M2-step is to make the semantic channel match the Shannon channel, the E-step is to make the Shannon channel match the semantic channel, and the M1-step is to make the destination *P*(*y*) match the source *P*(*x*). Repeating the above three steps can make the mixture model converge.

However, there are two problems with the EM algorithm: (1) *P*(*y*) may converge slowly; (2) if the likelihood functions are also fixed, how do we solve *P*(*y*)?

Based on the *R*(*G*) function analysis, the authors improved the EM algorithm to the EnM algorithm [25]. The E-step in the EnM algorithm remains unchanged, and the M-step is the M2-step in the EM algorithm. In addition, the n-step is added after the E-step. It repeats Equations (21) and (22) to calculate *P*(*y*) *n* times so that *P*^+1^(*y*) ≈ *P*(*y*). Both EM and EnM algorithms use the MIE criterion. The n-step can speed up *P*(*y*) matching *P*(*x*). The M-step only optimizes likelihood functions. Because after n-step, *P*(*y*_j_)/*P*^+1^(*y*_j_) is close to 1, we can use the following formula to optimize the model parameters:(24)$$P(x|{\theta_{j}}^{+1})=P(x)P(x|\theta_{j})/P_{\theta }(x).$$

Without the n-step, there will be *P*(*y_j_*) ≠ *P*^+1^(*y_j_*), and ∑*_i_ P*(*x_i_*)*P*(*x*|*θ_j_*)/*P_θ_*(*x_i_*) ≠ 1.

When solving the mixture model, we can choose a smaller *n*, such as *n*=3. When solving *P*(*y*) specifically, we can select a larger *n* until *P*(*y*) converges. When *n* = 1, the EnM algorithm reduces to the EM algorithm.

We can deduce that after the E-step, there is (see Appendix C for the proof):(25)$$KL(P||P_{\theta })=R-G+KL({P_{Y}}^{+1}\left|\right|P_{Y}),$$
where *KL*(*P*‖*P_θ_*) is the relative entropy or KL divergence between *P*(*x*) and *P_θ_*(*x*); *KL*(*P_Y_*^+1^‖*P_Y_*) is *KL*(*P*^+1^(*y*))‖*P*(*y*)), which is close to 0 after the M1 step or the n-step.

We can use Equation (25) to prove the convergence of mixture models. Since the M2-step maximizes *G*, and the E-step and n-step minimize *R* − *G* and *KL*(*P_Y_*^+1^‖*P_Y_*), *H*(*P*‖*P_θ_*) can be close to 0. Experiments have shown that, as long as the sample size is large enough, Gaussian mixture models with *n* = 2 will converge globally.

If the likelihood functions are fixed, the EnM algorithm becomes the En algorithm. The En algorithm can be used to find the latent variable *P*(*y*) for fixed constraint functions.

### 4.2. Goal-Oriented Information and Active Inference (About Sheep Herding)

In the above, we use the *G* measure to measure semantic information in communication systems, requiring that the prediction *P*(*x*|*θ_j_*) conforms to the fact *P*(*x*|*y_j_*). Goal-oriented information is the opposite, requiring the fact to conform to the goal. We also refer to this information as control information or purposeful information [34].

An imperative sentence can be regarded as a control instruction. We need to determine whether the control result aligns with the control goal. The more consistent the result is, the more information there is. A likelihood function or truth function can represent a control goal. The following goals can be expressed as truth functions:“The grain production should be close to or exceed 7500 kg/hectare”;“The age of death of the population should preferably exceed 80 years old”;“The cruising range of electric vehicles should preferably exceed 500 km”;“The error of train arrival time should preferably not exceed 1 min”.

Semantic KL information can be used to measure purposeful information:(26)$$I(X;a_{j}/\theta_{j})=\sum_{i}P(x_{i}|a_{j})\log\frac{T(\theta_{j}|x_{i})}{T(\theta_{j})},$$

In this formula, *θ_j_* indicates that the control target is a fuzzy range, and *a_j_* is an action selected for task *y_j_*. In the above formula, *y_j_* is replaced with *a_j_*. One reason is that we may select different *a_j_* for the same task *y_j_*; another reason is that *a_j_* is used in the popular active inference method for the same purpose.

If there are several control targets *y*_1_, *y*_2_, …, we can use the semantic MI formula to express the purposeful information:(27)$$I(X;A/\theta )=\sum_{j}P(a_{j})\sum_{i}P(x_{i}|a_{j})\log\frac{T(\theta_{j}|x_{i})}{T(\theta_{j})},$$
where *A* is a random variable taking a value *a_j_*. Using SVB, the control ratio *P*(*a*) can be optimized to minimize the control complexity (i.e., Shannon MI) for given *I*(*X*; *A*/*θ*).

Goal-oriented information can be regarded as the cumulative reward function in reinforcement learning. However, the goal here is a fuzzy range that represents a plan, command, or imperative sentence. The optimization task is similar to the active inference task using the FEP. For a multi-target task, the objective function to be minimized is*f* = *I*(*X*; *A*) − *sI*(*X*; *A*/*θ*).(28)

When the actual distribution *P*(*x*|*a_j_*) is close to *P*(*x*|*θ_j_*), the information efficiency reaches its maximum value of 1. To further increase both information, we can use the MID iteration with *s* to get *P*(*a_j_*|*x*) and *P*(*a*), and use Bayes’ formula to obtain(29)$$P*(x_{i}|a_{j})=P(a_{j}|x_{i})P(x_{i})/P(a_{j})=P(x_{i})m_{ij}^{s}/\sum_{k}P(x_{k})m_{kj}^{s}.$$

We may use the likelihood function *P*(*x*|*θ_j_*) or the truth function *T*(*θ_j_*|*x*) to represent goals for the sheep-herding task. We may modify Equation (29) to accommodate different constraint functions (see Equations (19) and (20)). Compared with VB, the above method is simpler, and we can change the constraint strength by *s*.

For both communication and constraint control (or active inference), *G* stands for the effect, *R* for the cost, and *G*/*R* for the efficiency. We collectively refer to *G*/*R* in both cases as information efficiency.

### 4.3. Comparison of R(G), Mixture Models, Active Inference, VB, and SVB

Table 1 shows that different tasks (in the first line) include different operations (in each column).

SVB encompasses all operations for minimizing *R* − *G* under specific constraints; every task or method must optimize *P(y*) and *P(y*|*x*) to minimize *R* or *R* − *G*.

## 5. The Relationship Between Information and, Physical Entropy and, Free Energy

### 5.1. Entropy, Information, and Semantic Information in Local Non-Equilibrium and Equilibrium Thermodynamic Systems

Gibbs set up the relationship between thermodynamic entropy and Shannon entropy; Jaynes [20,21] proved that, according to Stirling’s formula, ln*N*! = *N*ln*N − N* (when *N*→∞), there is a simple connection between Boltzmann’s microscopic state number *Ω* of *N* molecules and Shannon entropy:(30)$$S=k\ln\Omega =k\ln\frac{N!}{\prod_{i=1}^{G_{m}}N_{i}!}=-kN\sum_{i=1}^{G_{m}}P(x_{i}|T_{0})\ln P(x_{i}|T_{0})=kNH(X|T_{0}),$$
where *S* is entropy, k is the Boltzmann constant, *x_i_* is the *i*-th microscopic state (*i* = 1, 2, …, *G_m_*; *G_m_* is the microscopic state number of one molecule), *N* is the number of mutually independent molecules, *N_i_ i*s the number of molecules with *x_i_*, and *T*_0_ is the absolute temperature. *P*(*x_i_*|*T*_0_) = *N_i_*/*N* represents the probability of a molecule in a state *x_i_* at temperature *T*_0_. The Boltzmann distribution for a given energy constraint is as follows:(31)$$P(x_{i}|T_{0})=\exp(-\frac{e_{i}}{kT_{0}})/Z^{\prime },\;Z^{\prime }=\sum_{i}\exp(-\frac{e_{i}}{kT_{0}}),$$
where *Z*′ is the partition function.

Information and entropy in thermodynamic systems have been discussed by researchers [15,46], but the methods and conclusions in this paper differ.

Considering the information between temperature and molecular energy, we use Maxwell–Boltzmann statistics [47] (refer to Figure 5). Now, *x_i_* becomes energy *e_i_* (*i* = 1, 2, …, *m*); *g_i_* stands for the microscopic state number of a molecule with energy *e_i_* (i.e., degeneracy), and *N_i_* for the number of molecules with energy *e_i_*.

According to the classical probability definition, the prior probability of each microscopic state of a molecule is *P*(*x_i_*) = 1/*G_m_;* the prior probability of a molecule with energy *e_i_* is *P*(*e_i_*) = *g_i_*/*G_m_*. The posterior probability is *P*(*e_i_*|*T*_0_) = *N_i_*/*N.* So, Equation (30) becomes
(32)$$\begin{array}{l} S=k\ln(N!\prod_{i=1}^{m}\frac{{g_{i}}^{N_{i}}}{N_{i}!})=-kN\sum_{i=1}^{m}P(e_{i}|T_{0})\ln\frac{P(e_{i}|T_{0})}{g_{i}}\\ \;\;=-kN\sum_{i}P(e_{i}|T_{0})\ln\frac{P(e_{i}|T_{0})}{P(e_{i})}+kN\ln G=kN[\ln G-KL(P(e|T_{0})||P(e)]. \end{array}$$
Under the energy constraint, when the system reaches equilibrium, Equation (31) becomes(33)$$P(e_{i}|T)=P(e_{i})\exp(-\frac{e_{i}}{kT})/Z,\;Z=\sum_{i}P(e_{i})\exp(-\frac{e_{i}}{kT}).$$

Consider a local non-equilibrium system. Different regions *y_j_* (*j* = 1, 2, …) of the system have different temperatures *T_j_* (*j* = 1, 2, …). Hence, we have *P*(*y_j_*) = *P*(*T_j_*) = *N_j_*/*N* and *P*(*x*|*y_j_*) = *P*(*x*|*T_j_*). From Equation (32), we obtain(34)$$\begin{array}{ll} I(E;Y) & =\sum_{j}P(y_{j})KL(P(e_{i}|y_{j})||P(e_{i}))=\sum_{j}P(y_{j})[\ln G_{m}-S_{j}/(kN_{j})]\\ & \;=\ln G_{m}-S(kN), \end{array}.$$
where ln*G_m_* is the prior entropy *H*(*X*) of *X*. Let *E* be a random variable taking a value *e*. Since *e* is certain for a given *x*, there are *H*(*E*|*X*) = 0 and *H*(*X*, *E*) = *H*(*X*) = ln*G_m_*. From Equation (34), we derive(35)$$I(E;Y)=\ln G_{m}-S/(kN)=H(X)-H(X|Y)=I(X;Y).$$
This formula indicates that the information about energy *E* provided by *Y* is equal to the information about microscopic state *X*, and *S*/(*kN*) *= H*(*X*|*Y*).

According to Equations (31)–(33), when local equilibrium is reached, there is(36)$$\begin{array}{l} \;I(X;Y)=I(E;Y)=\sum_{j}\sum_{i}P(e_{i},y_{j})\ln\frac{\exp[-e_{i}/(kT_{j})]}{Z_{j}}\\ =-\sum_{j}P(y_{j})\log Z_{j}-E(e/T)/(kN)=H(Y_{\theta })-H(Y_{\theta }|X)=I(X;Y_{\theta }), \end{array}$$
where E(*e*/*T*) is the average of *e*/*T*, which is similar to relative square error. It can be seen that in local equilibrium systems, minimum Shannon MI can be expressed by the semantic MI formula. Since *H*(*X*|*Y*) becomes *H*(*X*|*Y_θ_*), there is(37)$$\;S=kNH(X|Y_{\theta })=kNF,$$
which means that VFE (*F*) is proportional to thermodynamic entropy.

Why do physics and G theory have the same forms of entropy and information? It turns out that the entropy and information in both are subject to certain constraints. In physics, it is the energy constraint, while in G theory, it is the extension constraint. There is a simple connection between the two: *T*(*θ_j_*|*x_i_*) = exp[−*e_i_*/(k*T_j_*)].

### 5.2. Information, Free Energy, Work Efficiency, and Information Efficiency

Helmholtz’s free energy formula is as follows:
*F*″ = *U* − *T*_0_*S*,(38)
where *F*″ is free energy, and *U* is the system’s internal energy. In an open system, the free energy may increase when the system changes from an equilibrium state to a non-equilibrium state. When *U* remains constant, there is(39)$$\;\Delta F\prime \prime =-\Delta (TS)=T_{0}S-\sum_{j}T_{j}S_{j}=kNT_{0}H(X|T_{0})-kN\sum_{j}T_{j}H(X|Y).$$
If *T*_0_ approaches ∞, *H*(*X*|*T*_0_) is close to *H*(*X*). Comparing the above equation with the Shannon MI formula, *I*(*X*;*Y*) *= H*(*X*) − *H*(*X*|*Y*), we can find that Shannon MI is like the increment of free energy in a local non-equilibrium system.

In thermodynamics, exergy is the energy that can do work [48]. When a system’s state is changed, exergy is defined as*Exergy* = (*E* − *E*_0_) + *p*_0_(*V* − *V*_0_) − *T*_0_(*S* − *S*_0_),(40)
where *E − E*_0_ is the increment of the system’s internal energy, *p*_0_ and *T*_0_ are the pressure and temperature of the environment, and *V* − *V*_0_ is the increment of volume. In local non-equilibrium systems, if the volume and temperature of each local region are constant,Δ*Exergy* < Δ*F*″.(41)
When the system reaches local equilibrium, there isΔ*Exergy* = Δ*F*″.(42)
It can be seen that in a local equilibrium system, semantic MI is equivalent to the increment of free energy, which is equivalent to the increment of exergy. Semantic MI is less than or equal to Shannon MI, just as exergy is less than or equal to free energy *F*″.

We can also regard k*NT*_0_ and k*NT_j_* as the unit information values [23], so Δ*Exergy* is equivalent to the increase in information value.

Generally speaking, the larger the free energy, the better. Only when free energy is used to do work do we want to consume less free energy. Similarly, we want to consume less Shannon information only when it is used to convey semantic information. Semantic information *G* parallels work *W*. The ratio *W*/Δ*F*″ reflects work efficiency; similarly, the ratio *G*/*R* reflects information efficiency.

## 6. The FEP and Inconsistency Between Theory and Practice

### 6.1. VFE as the Objective Function and VB for Solving Latent Variables

Hinton and Camp [1] provided the following formula:(43)$$F=\sum_{j}r_{j}E_{j}-\sum_{j}r_{j}\log\frac{1}{r_{j}},$$
where *r_j_* is the above *P*(*y*_j_), and *E_j_* is the encoding cost of *x* according to *y_j_*, which is also called the reconstruction cost. *F* is called “free energy” because the formula is similar in form to the free energy formula in physics. In thermodynamics, minimizing free energy can obtain the Boltzmann distribution. Similarly, we can get(44)$$\begin{array}{l} \;r_{j}=\exp(-E_{j})/\sum_{j}\exp(-E_{j}),\\ E_{j}=H(X,\theta_{j})=-\sum_{i}P(x_{i}|y_{j})\log P(x_{i},y_{j}|\theta ). \end{array}$$
Hinton and Camp’s variation method was developed into a more general VB method. The objective function of VB is usually expressed as follows [5]:(45)$$F=\sum_{y}g(y)\log\frac{g(y)}{P(x,y|\theta )}=-\sum_{y}g(y)\log P(x|y,\theta )+KL(g(y)||P(y)).$$
where *g*(*y*) is *P*^+1^(*y*) in SVB. Negative *F* is usually called the evidence lower bound, denoted by L(*g*). In the above formula, *x* should be a vector and related to *y*. Using the semantic information method, we express *F* as(46)$$\begin{array}{ll} F & =\sum_{i}P(x_{i})\sum_{i}P(y_{j}|x_{i})\log\frac{P^{+1}(y_{j})}{P(x_{i},y_{j}|\theta )}\\ & \;=\;\sum_{j}P^{+1}(y_{j})\sum_{i}P(x_{i}|y_{j})\log\frac{P^{+1}(y_{j})}{P(x_{i}|\theta_{j})P(y_{j})}=H(X|Y_{\theta })+KL({P_{Y}}^{+1}||P_{Y}). \end{array}$$

This formula can better describe how the latent variable *P(y*) fits observed data *P(x*) than Equation (45). In the EM algorithm, after the M1-step or after iteration convergence, *P*^+1^(*y*) equals *P*(*y*), and hence *F* equals *H*(*X*|*Y_θ_*). So, the author says, “roughly speaking, *F* = *H*(*X*|*Y_θ_*).” Hereafter, we assume that *F* is calculated after the M1-step, so *F = H*(*X*|*Y_θ_*). Hence, the relationship between the information difference *R* − *G* and *F* is*R − G* = *I*(*X*; *Y*) − *I*(*X*; *Y_θ_*) = *H*(*X*|*Y_θ_*) − *H*(*X*|*Y*) = *F* − *H*(*X*|*Y*).(47)

To optimize *P*(*y*), the mean-field approximation [5] is often used to optimize *P*(*y*|*x*) first and then obtain *P*(*y*) from *P*(*y*|*x*) and *P*(*x*). This is to use *P*(*y*|*x*) as the variation to minimize(48)$$F^{\#}=\sum_{x}P(x)\sum_{y}P(y|x)\log\frac{P(y|x)}{P(x,y|\theta )}.$$

Minimizing *F*^#^ is equivalent to minimizing the cross-entropy:(49)$$H_{\theta }(X)=-\sum_{i}P(x_{i})\log P_{\theta }(x_{i}).$$
It is also equivalent to minimizing *KL*(*P*||*P_θ_*) and *R − G*, which can make the mixture model converge. Therefore, the MIE criterion is also employed.

Neal and Hinton used the VB to improve the EM algorithm [2] to solve the mixture model. They defined*F*(*P*(*y*), *θ*) = E_*P*(*x*,*y*)_ log *P*(*x*, *y*|*θ*) + *H*(*Y*)(50)
as negative VFE and maximize *F*′ = *F*(*P*(*y*), *θ*) = −*F* for the convergence of mixture models.

Neal and Hinton showed that using the Incremental Algorithm (see Equation (7) in [2]) to maximize *F*′ in both the E-step and the M-step can make mixture models converge faster. Their M-step is the same as the M-step in the EnM algorithm, but the E-step only updates one *P*(*y*_j_|*x*) each time, leaving *P*(*y_k_*|*x*)(*k* ≠ *j*) unchanged, similar to the mean field approximation used by Beal [3]. They actually also minimized *H_θ_*(*X*), *KL*(*P*||*P_θ_*), and *R* − *G*.

Experiments show that the EM algorithm, the EnM algorithm, the Incremental Algorithm, and the VB-EM algorithm [3] can all make mixture models converge, but unfortunately, during the convergence, *F*′ may not continue to increase, or *F* may not continue to decrease, which means that the calculation results of VB are correct, but the theory is imperfect.

Some people use the continuous increase in the complete data log-likelihood *Q* = −*H*(*X, Y_θ_*) to explain or prove the convergence of the EM algorithm [38,39]. However, *Q*, like *F*′, may also decrease during the convergence of mixture models.

### 6.2. The Minimum Free Energy Principle

Friston et al. first applied VB to brain science and later extended it to biobehavioral science, thus developing the MFE criterion into the FEP [6,7].

The FEP uses *μ*, *a*, *s*, and *η* to represent four states, respectively:*μ*: internal state, which parametrizes the approximate posterior distribution or variational density over external states; in SVB, it is the likelihood function *P*(*x*|*θ_j_*) to be optimized by the sampling distribution.*a*: subjective action; in SVB, it is *y* (for prediction) and *a* (for constraint control).*s*: perception; that is, the above observed datum *x*.*η*: external state, which is objective *y* and *P*(*x*|*y*); when SVB is used for constraint control, it is the external response *P*(*x*|*β_j_*), which is expected to be equal to *P*(*x*|*θ_j_*).

According to the FEP, there are two types of optimization [6]:(51)$$\begin{array}{l} \mu *=\underset{\mu }{\arg\min}\{F(\mu ,a;s)\},\\ a*=\underset{a}{\arg\min}\{F(\mu *,a;s)\}. \end{array}$$

The first equation optimizes the likelihood function *P*(*s*|*μ*). The second equation optimizes the action selection: *P*(*a*) and *P*(*a*|*s*). The latter is also called active inference. This is similar to finding *P*(*a*) and *P*(*a*|*x*) when using SVB for constraint control.

Friston sometimes interprets *F* as unexpectedness, surprise, and uncertainty and sometimes as error. Reducing *F* means reducing surprise and error. This is easy to understand from the perspective of G theory. Because *F* is the semantic conditional entropy, reflecting the average code length of residual coding after the prediction. When the event conforms to the prediction or purpose, uncertainty and surprise are reduced, and semantic information increases. Reducing *F* means reducing the error because there is *G* = *H_θ_*(*Y*) − $\bar{d}$ = *H*(*X*) *− F*; when *H_θ_*(*Y*) and *H*(*X*) are fixed, *F* and $\bar{d}$ increase or decrease at the same time.

### 6.3. The Progress from the ME Principle to the FEP

In 1957, Jaynes proposed the ME principle [20,21]. He regards physical entropy as a special case of information entropy and provides a method for solving ME distributions. This method can be used to predict the probability distribution of a system state under specific constraints and to optimize the control of random events within these constraints. Compared with the Entropy Increase Law, the ME principle uses active constraint control, enabling its application to human intervention in nature. However, bio-organisms have purposes, and the ME principle cannot evaluate whether their predictions and controls align with these facts or purposes.

According to the FEP, the smaller the *F*, the more the subjective prediction *P*(*x*|*θ_j_*) conforms to the objective fact *P*(*x*|*y_j_*). On the other hand, the active inference with the FEP makes the objective fact *P*(*x*|*y_j_*) closer to the subjective purpose *P*(*x*|*θ_j_*). Both principles maximize the Shannon conditional entropy *H*(*X*|*Y*). However, the FEP also optimizes the objective function using the maximum likelihood criterion, a common approach in machine learning. Therefore, compared to the ME principle, the FEP can better optimize the prediction and adaptation (including intervention) of the bio-organism to its environment, promoting self-organization and resisting the increase in Earth’s entropy.

### 6.4. Why May VFE Increase During the Convergence of Mixture Models?

The author considered the properties of cross-entropy(52)$$H(X|\theta_{j})=-\sum_{i}P(x_{i}|y_{j})\log P(x_{i}|\theta_{j})$$
a long time ago. When *P*(*x*|*θ_j_*) approaches a fixed *P*(*x*|*y_j_*), *H*(*X*|*θ_j_*) decreases. But conversely, when *P*(*x*|*y_j_*) approaches a fixed *P*(*x*|*θ_j_*), will *I*(*X*; *θ_j_*) increase or decrease? It is uncertain. What is certain is that *KL*(*P*(*x*|*y_j_*)||*P*(*x*|*θ_j_*)) = *I*(*X*; *y_j_*) − *I*(*X*; *θ_j_*) will decrease. For example, *P*(*x*|*y_j_*) and *P*(*x*|*θ_j_*) are two Gaussian distributions with the same expectation. And the standard deviation *d*_1_ of *P*(*x*|*y_j_*) is smaller than the standard deviation *d*_2_ of *P*(*x*|*θ_j_*). The *d*_1_ will increase while *P*(*x*|*y_j_*) approaches *P*(*x*|*θ_j_*), and the cross-entropy will also increase.

Table 2 shows a simpler example, where *x* has four possible values. When *P*(*x*|*y_j_*) changes from the concentrated to the dispersed, *H*(*X*|*θ_j_*) will increase.

This conclusion can be extended to the semantic MI formula, concluding that when the Shannon channel matches the semantic channel, *I*(*X*; *Y_θ_*) and *H*(*X*|*Y_θ_*) = *F* are uncertain to increase or decrease; what is certain is that *R* − *G* must decrease. VB also ensures that the Shannon channel matches the semantic channel, so during the matching process, *R* − *G* = *F* − *H*(*X*|*Y*) instead of *F* will continue to decrease.

During the iteration of Gaussian mixture models, two causes affect the increase or decrease in *F* and *H*(*X*|*Y_θ_*):

(1) At the beginning of the iteration, the distribution range of *P*(*x*|*θ_j_*) and *P*(*x*|*y_j_*) is quite different. After *P*(*x*|*θ_j_*) approaches *P*(*x*|*y_j_*), *F* and *H*(*X*|*Y_θ_*) will decrease.

(2) The true model’s VFE *F*=*H*(*X*|*Y*) (i.e., Shannon conditional entropy) is very large. During the iteration, *F* and *H*(*X*|*Y_θ_*) may increase.

When reason (1) is dominant, *F* decreases; when reason (2) is dominant, *F* increases.

Suppose a Gaussian mixture model has two components; only two initial standard deviations are smaller than the true model’s two standard deviations. During the iteration, *F* will continue to increase (see Section 7.1.2).

Asymmetric standard deviations and mixing ratios can also cause *F* to increase sometimes (see Section 7.1.1). In addition, the initial parameters *µ*_1_ and *µ*_2_ are biased towards one side, which may cause *F* increase in some steps (see Section 7.1.3).

## 7. Experimental Results

### 7.1. Proving Information Difference Monotonically Decreases During the Convergence of Mixture Models Rather than VFE

#### 7.1.1. Neal and Hinton’s Example: Mixture Ratios Cause *F*′ and *Q* to Decrease

According to the popular view, during the convergence of mixture models, *F*′ = −*H*(*X*|*Y_θ_*) = −*F* and *Q* = −*H*(*X*, *Y_θ_*) continue to increase. However, counterexamples are often seen in experiments. First, let us look at the example of Neal and Hinton [2] (see Table 3 and Figure 6). Table 3 shows the true and initial model parameters and the mixture ratios (the values of *x* below are magnified, and the magnified formula is *x* = 20(*x*′ − 50) (*x*′ is the original value in [2]).

After the true model’s mixture ratio was changed from 0.7:0.3 to 0.3:0.7, *F*′ did not always increase. The reason was that the cross-entropy *H*(*X*|*y*_2_) of the second component of the true model was relatively large. So *H*(*X*|*Y_θ_*) increased with *P*(*y*_2_). Later, *F*′ eventually increased because the cross-entropy decreased after *P*(*x*|*θ_j_*) approached *P*(*x*|*y_j_*).

#### 7.1.2. A Typical Counterexample Against VB and the FEP

Table 4 and Figure 7 illustrate a mixture model where the initial two standard deviations are smaller than those of the true model. During the iteration process, *F*′ and *Q* continued to decrease (except at the beginning).

#### 7.1.3. A Mixture Model with Poor Convergence

Figure 8 shows an example with poor convergence from [39]. The true model parameters are (*µ*_1_*, *µ*_2_*, *σ*_1_*, *σ*_2_*, *P**(*y*_1_)) = (100, 125, 10, 10, 0.7). To make convergence more difficult, we set the initial model parameters to (*µ*_1_, *µ*_2_, *σ*_1_, *σ*_2_, *P*(*y*_1_)) = (80, 95, 5, 5, 0.5). Experiments showed that as long as the sample was large enough, the EM algorithm, the EnM algorithm, the Incremental Algorithm [2], and the VBEM algorithm [3] could all converge. However, during the convergence process, only *R–G* and *KL*(*P*||*P_θ_*) continued to decrease, while *F*′ and *Q* did not continue to increase. This example indicates that if the initialization of *µ*_1_ and *µ*_2_ is inappropriate, *F*′ and *Q* may also decrease during the iteration. The decrease in *Q* is more obvious.

This example also demonstrates that the E3M algorithm requires fewer iterations (240 iterations) than the EM algorithm (350 iterations).

### 7.2. Simplified SVB (The En Algorithm) for Data Compression

The task was that 8-bit grayscale pixels (256 gray levels) were compressed into 3-bit pixels (8 gray levels). Considering that the eye’s grayscale discrimination is higher when the brightness is low, we used eight truth functions shown in Figure 9a as the constraint functions. Given *P*(*x*) and *T*(*y*|*x*), we found the Shannon channel *P*(*y*|*x*) for MIE.

*P*(*y*) converged after repeating the MID iteration three times. At the beginning of the iteration, it was assumed that *P*(*y*_j_) = 1/8 (*j* = 0, 1, …), and the entropy *H*(*Y*) was 3 bits. When the iteration converged, *R* was 2.299 bits, *G* was 2.274 bits, and *G*/*R* was 0.989. These results mean that a 3-bit pixel can be transmitted with about 2.3 bits.

### 7.3. Experimental Results of Constraint Control (Active Inference)

We simplified the sheep-herding space into a one-dimensional space with only two pastures (see Figure 10) to illustrate the relationship between the control results and the goals (the constraint ranges), as well as how the control results changed with *s*. We needed to infer the latent variable *P*(*a*) given *P*(*x*) and the two truth functions.

For different *s*, we set the initial ratios: *P*(*a*_0_) = *P*(*a*_1_) = 0.5. Then, we used the MID iteration to obtain optimal *P*(*a_j_*|*x*) (*j* = 0, 1). Then, we got *P*(*x*|*a_j_*) = *P*(*x*|*θ_j_*, *s*) by(53)$$P(x_{i}|a_{j},s)=P(a_{j}|x_{i},s)P(x_{i})/P(a_{j})=P(x_{i})m_{ij}^{s}/\sum_{k}P(x_{k})m_{kj}^{s}.$$
Then, we used the parameter solution of the *R*(*G*) function to obtain *G*(*s*), *R*(*s*), and *R*(*G*(*s*)). Figure 10a,b show *P*(*x*|*θ_j_*, *s*) and *P*(*x*|*β_j_*, *s*) for *s* = 1 and *s* = 5, respectively. When *s* = 5, the constraints are stricter, and some sheep at fuzzy boundaries are moved to more ideal positions. Figure 11 shows that when *s* > 5, *G* changes very little, indicating that we need to balance maximum purposeful information *G* and the MIE *G*/*R*. A larger *s* will reduce information efficiency and is unnecessary.

The dashed line for *R*_1_(*G*) indicates that if we replace *P*(*x*|*a_j_*) = *P*(*x*|*θ_j_*, *s*) with a normal distribution, *P*(*x*|*β_j_*, *s*), *G* and *G*/*R*_1_ do not obviously become worse.

For the constraint control of one task (*Y* = *y_j_*), such as for the age control of death of adults by medical conditions, we can set *R* = *I*(*X*; *y_j_*), *G* = *I*(*X*; *θ_j_*). The optimization method is similar, but there is no need to find latent variables. For details, see [34].

## 8. Discussion

### 8.1. Three Defects in the FEP and Remedies

Based on the previous analysis and experimental results, we identified three defects in VB and the FEP.

The first defect is that its convergence theory is flawed. Although the computed results are correct, VFE does not always decrease monotonically during the convergence of mixture models. As demonstrated in Section 7.1 (see Figure 6, Figure 7 and Figure 8), only the information difference *R* − *G* consistently decreases.

The second defect is that the term “free energy” is easily misunderstood. As discussed in Section 5.1, *H*(*X*∣*Y_θ_*) = *F* is proportional to thermodynamical conditional entropy in a local equilibrium system. Consequently, interpreting *H*(*X*||*Y_θ_*) as free energy can lead to a misunderstanding.

The third defect is that the FRP has a limitation: it only utilizes likelihood functions as constraints without incorporating range or semantic constraints.

To remedy the first defect, we employ SVB. Section 3.3 and Section 3.4 show that SVB is theoretically and practically consistent. The experiments in Section 7.1 demonstrate that throughout the iterative process of estimating latent variables, the information difference *R* − *G*, instead of VFE, continuously decreases, or the information efficiency *G*/*R* steadily increases.

To remedy the second defect, we use the MIE criterion instead of the MFE criterion, allowing us to interpret that bio-organisms, particularly humans, can promote the Earth’s order by acquiring more information and preserving more free energy.

To address the third defect, SVB, based on the P-T probability framework, also incorporates truth, membership, similarity, and distortion functions as constraints. These functions represent semantics and are used to define the semantic information measure, which more effectively assesses the conformity between the subjective prediction (or purposes) and the objective reality when bio-organisms predict and regulate their environments.

### 8.2. Similarities and Differences Between SVB and VB

Both SVB and VB aim to perform two fundamental tasks:

(1) Optimizing model parameters or likelihood functions.

(2) Using variational methods to infer latent variables *P*(*y*) according to observed data and constraints.

However, they differ in several key aspects:**Optimization Criteria**: Both VB and SVB optimize model parameters using the maximum likelihood criterion. When optimizing *P*(*y*), VB nominally follows the MFE criterion but, in practice, employs the minimum KL divergence criterion (i.e., minimizing KL(*P*∣∣*P_θ_*) to make mixture models converge). This criterion is equivalent to the MIE criterion used in SVB.**Variational Methods**: VB uses either *P*(*y*) or *P*(*y*∣*x*) as the variation, whereas SVB alternatively uses *P*(*y*∣*x*) and *P*(y) as the variations.**Computational Complexity**: VB relies on logarithmic and exponential functions to compute *P*(*y*∣*x*) [3,5], leading to relatively high computational complexity. In contrast, SVB offers a simpler approach to calculate *P*(*y*∣*x*) and *P*(*y*) for the same task (*s* = 1).**Constraint Functions**: VB can only use likelihood functions as constraints. In contrast, SVB allows for various functions, including likelihood and truth functions. In addition, the constraint in SVB can be enhanced by the parameter *s* (see Figure 10 and Figure 11).

SVB is potentially more suitable for various machine learning applications. However, because SVB does not consider the probability of the parameters, it may not be as applicable as VB in certain situations.

### 8.3. Optimizing the Shannon Channel with the MFE or MIE Criterion for Communication

Shannon information theory uses the distortion criterion instead of the information criterion when optimizing channels for data compression. Hinton and Camp [1] initially employed the MFE criterion to compress data, which is consistent with minimizing the residual coding length. VB’s success in the field of machine learning suggests that VFE or *H(X*|*Y_θ_*) is a more suitable optimization measure for optimizing Shannon channels than distortion for electronic communication.

SVB uses the MID or MIE criterion, which is essentially the same as the MFE criterion. The MIE criterion is easier to understand and apply. In addition, SVB enables us to utilize various functions as constraints to optimize the Shannon channel (see Section 7.2).

### 8.4. Two Directions Worth Exploring

#### 8.4.1. Using High-Dimensional Truth Functions to Express VFE or H(*X*|*Y_θ_*)

Using truth functions as learning functions to produce likelihood functions and express VFE or semantic conditional entropy *H*(*X*|*Y_θ_*) has the following advantages [25]:When *P*(*x*) is changed, the optimized truth function is still valid.The transit probability function *P*(*y_j_*|*x*) is often approximately proportional to a Gaussian function (its maximum is 1) rather than the Gaussian distribution (its sum is 1), whereas *P*(*x*|*y_j_*) is not. In these cases, it is better to use the Gaussian function rather than the Gaussian distribution as the learning function, obtaining *T**(*θ_j_*|*x*) ∝ *P*(*y_j_*|*x*).
The GPS pointer is a simple example [25]. We can learn the semantics (i.e., a Gaussian function) of a GPS pointer from a sample {(*x*(*t*), *y*(*t*), *t* = 1, 2, …}, where *x* represents a real position and *y* represents a pointed position. In this example, *P*(*x*) and *P*(*x*|*y_j_*) are not necessarily Gaussian distributions.

However, the author has thus far only used one-dimensional and two-dimensional Gaussian functions as learning functions. In contrast, Friston et al. have used high-dimensional Gaussian distributions with the Hessian matrix for VFE. Now, we can combine the two methods to express VFE or semantic conditional entropy *H*(*X*|*Y_θ_*).

Sometimes, the forms of the truth function are unknown. We may use deep learning methods to find them. Word2Vec [49,50] is a highly successful deep learning method for capturing the complex semantics of words. It is worth exploring whether we can combine deep learning methods, including Word2Vec, with G theory to obtain better results.

#### 8.4.2. Trying to Explain Sensory Adaptability by Using the MIE Principle

The author has studied the evolution of color vision and preferences for beauty (see the author’s homepage, whose web address is in Appendix A). Inspired by Friston’s work, the author attempts to address some issues with biological evolution using the MIE principle.

Fruit- and nectar-feeding birds and humans can perceive various colors, while predators like eagles can only see black and white but are highly sensitive to motion. Why has evolution not produced animals with both capabilities? The MIE principle explains that Shannon information must match the information and information value that animals care about to produce high information efficiency.

According to behavioral biology, pleasure, including the sense of beauty, reinforces behaviors that are beneficial to survival. Yet substances like tobacco or drugs bring pleasure despite long-term harm. We can explain that an accurate assessment of pros and cons requires high information costs and results in low information efficiency. Of course, pleasure mechanisms are evolving and may also lag in adapting to novel environments.

Humans often find beauty in artificial images, just as they do in real, natural scenes. Similarly, peacocks that favor berries are especially drawn to tree patterns laden with berries. Using the MIE principle, we can explain that intuitive esthetic responses, which require lower information costs than rational analysis, offer higher information efficiency.

## 9. Conclusions

The FEP inherits the positive insight from EST that living systems increase the Earth’s order by self-organization. Unlike the ME principle, the FEP implicitly combines the maximum likelihood criterion (for optimizing model parameters) with the ME principle (for maximizing conditional entropy *H*(*X*|*Y*)). This theoretical framework explains how bio-organisms predict and adapt to their environments. Furthermore, the optimization technique may be applied to promote the sustainable development of ecological systems. However, there are three reasons for improving the FEP: it has a theoretical flaw, a possibility of being misunderstood, and a limitation. The author developed the semantic information G theory based on the P-T probability framework, which can overcome the limitation. This paper proposes SVB, which can more conveniently solve the mathematical problem that VB solves without that flaw.

To eliminate misunderstandings and provide a clearer explanation of biological self-organization, this paper proposes the MIE principle. It uses local non-equilibrium and equilibrium systems as examples to clarify the relationships among entropy, physical free energy, VFE, Shannon information, and semantic information. The main conclusions include that Shannon MI corresponds to the increment of free energy in a local non-equilibrium system; semantic MI corresponds to the increment of exergy or the increment of free energy in a local equilibrium system; VFE (*F*) represents physical (conditional) entropy in a local non-equilibrium system. Thus, we may say that acquiring and increasing information is similar to acquiring and increasing free energy; maximizing information efficiency parallels maximizing work efficiency. In this way, we can explain—without contradiction or misunderstanding—how biologically organized systems can increase the Earth’s order and counteract the growth of entropy by collecting and utilizing information and free energy. This explanation is also more compatible with EST.

While promising, the MIE principle is still evolving. It should build upon the foundational ideas of the FEP and explore integration with deep learning techniques to extend its applicability.

## Figures and Tables

**Figure 1 entropy-27-00684-f001:**
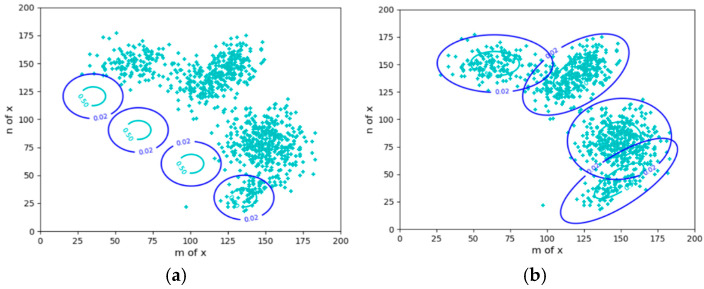
Explaining the Gaussian mixture model using sheep clustering as an example. (**a**) The iteration starts; (**b**) The iteration converges. The *x* = (*m*, *n*) is a two-dimensional point.

**Figure 2 entropy-27-00684-f002:**
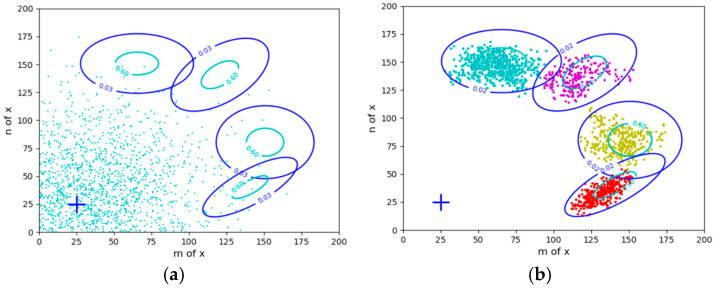
Taking herding sheep as an example to illustrate the constraint control of uncertain events (the constraint condition is some fuzzy ranges). (**a**) Control starts. (**b**) Control ends.

**Figure 3 entropy-27-00684-f003:**
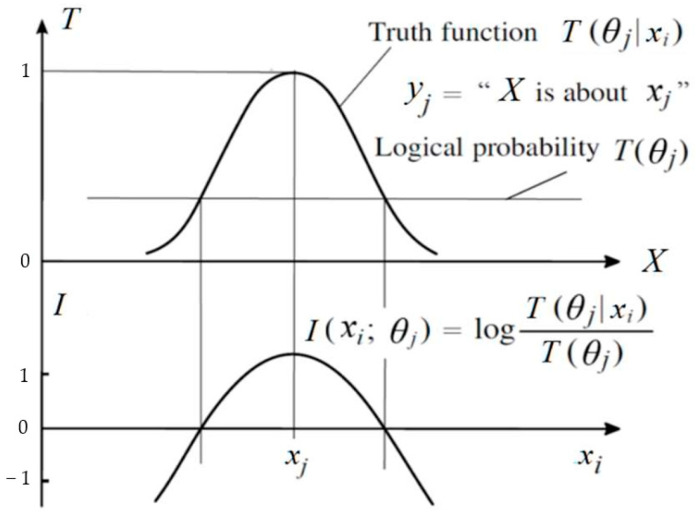
Illustrating the amount of semantic information *I*(*x_i_*; *θ_j_*). If *T*(*θ_j_*|*x_i_*) is larger and *T*(*θ_j_*) is smaller, then the *I*(*x_i_*; *θ_j_*) is larger. If the deviation is larger, the information may be negative.

**Figure 4 entropy-27-00684-f004:**
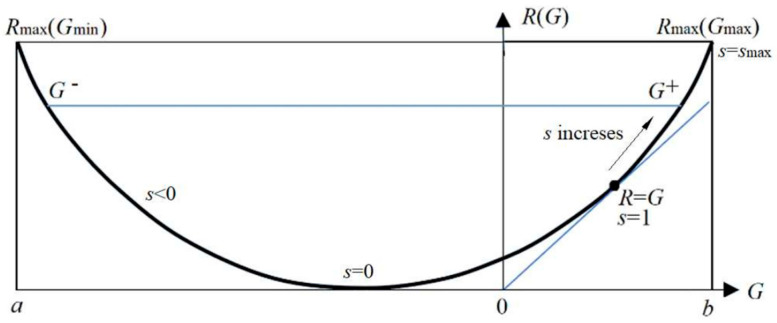
The information rate-fidelity function *R*(*G*) for binary communication. There is a point at which *R*(*G*) = *G* (*s* = 1). For given *R*, two anti-functions exist: *G*^−^(*R*) and *G*^+^(*R*).

**Figure 5 entropy-27-00684-f005:**
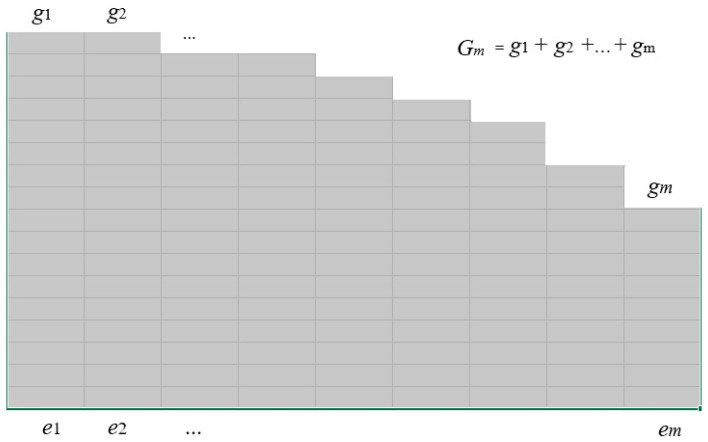
Degeneracy gi is the microstate number of a molecule with energy ei.

**Figure 6 entropy-27-00684-f006:**
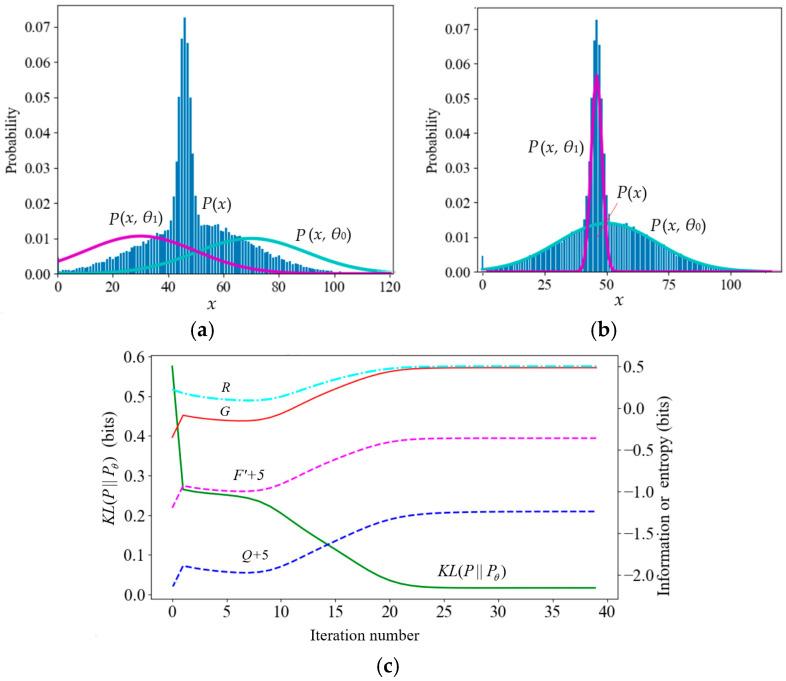
The convergence process of the mixture model that Neal and Hinton used. After the true model’s mixture ratio was changed from 0.7:0.3 to 0.3:0.7, *F*′ and *Q* decreased in some steps. (**a**) The iteration starts; (**b**) iteration converges; (**c**) *R*, *G*, *F*′, and *Q* change in the iteration process.

**Figure 7 entropy-27-00684-f007:**
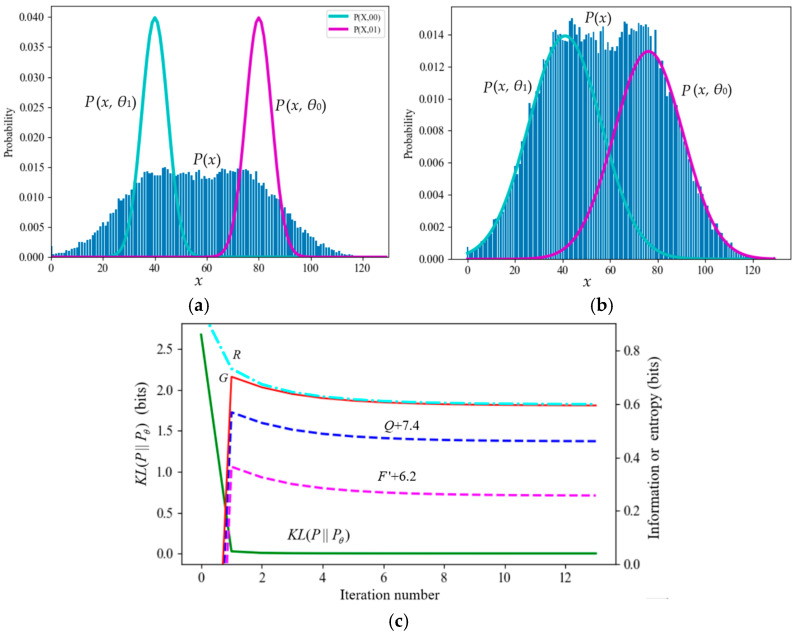
A typical mixture model against the FEP. (**a**) The iteration starts. (**b**) The iteration converges. (**c**) *R*, *G*, *F*′, and *Q* change in the iteration process.

**Figure 8 entropy-27-00684-f008:**
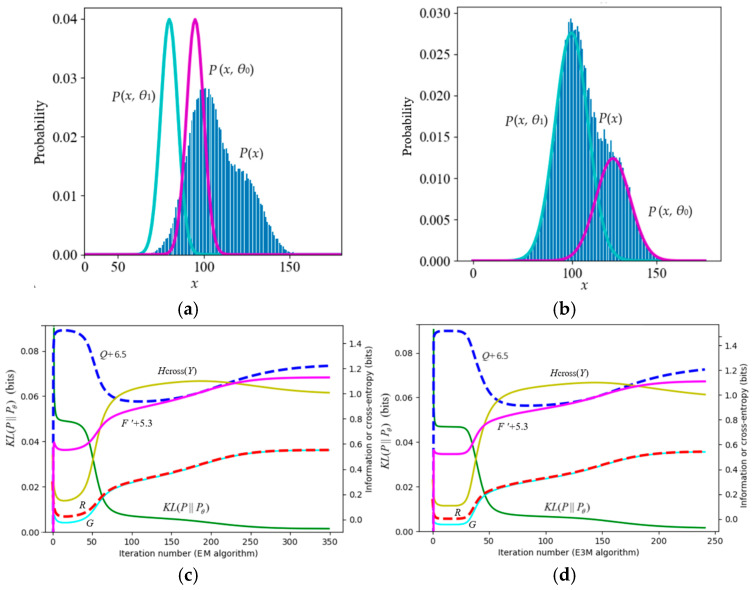
The mixture model with poor convergence. *H_cross_*(*Y*) = −∑*_j_ P*^+1^(*y_j_*)log*P*(*y_j_*) is the cross-entropy. (**a**) The iteration starts; (**b**) the iteration converges; (**c**) the iteration process of the EM algorithm; (**d**) the iterative process of the E3M algorithm.

**Figure 9 entropy-27-00684-f009:**
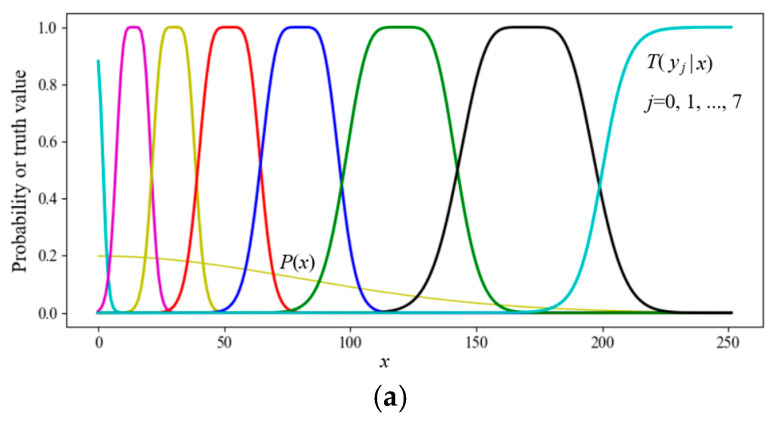
Using the En algorithm to optimize the Shannon channel *P*(*y*|*x*). (**a**) Eight truth functions as the constraint; (**b**) the optimized *P*(*y*|*x*); (**c**) *R, G* and *H*(*Y*) change with the MID iteration.

**Figure 10 entropy-27-00684-f010:**
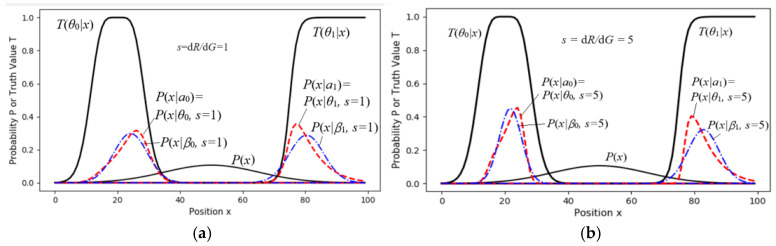
A two-objective control task. (**a**) For the case with *s* = 1; (**b**) for the case with *s* = 5. *P*(*x*|*β_j_*, *s*) is a normal distribution produced by the action *a_j_*.

**Figure 11 entropy-27-00684-f011:**
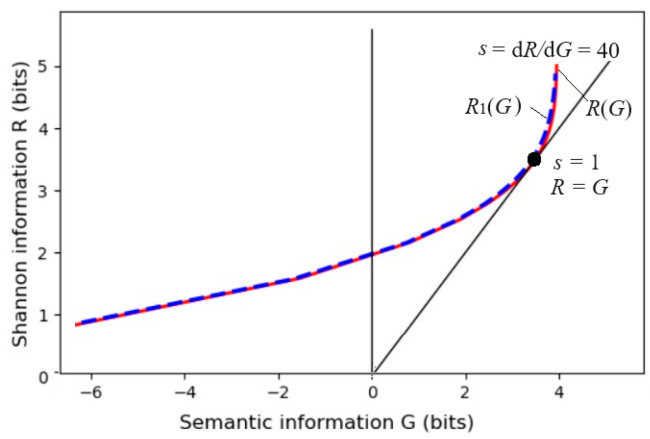
The *R*(*G*) for constraint control. *G* slightly increases when *s* increases from 5 to 40, meaning *s* = 5 is good enough.

**Table 1 entropy-27-00684-t001:** Different operations for different tasks or methods.

Task or Method	*R*(*G*)	Mixture Models	Active Inference	VB	SVB
Optimize *P*(*x*|*θ_j_*)	may	yes	may	yes	yes
Optimize *P*(*y*) and *P*(*y*|*x*)	yes	yes	yes	yes	yes
Use *s*	yes	no	may	not yet	yes
Use Constraint *T*(*y*|*x*)	yes	may	may	not yet	yes
Change *P*(*x*)	no	no	yes	yes	yes

**Table 2 entropy-27-00684-t002:** *H*(*X*|*θ_j_*) increases when *P*(*x*|*y_j_*) approaches *P*(*x*|*θ_j_*).

	*x* _1_	*x* _2_	*x* _3_	*x* _4_	*H*(*X*|*θ_j_*) (Bits)
*P*(*x*|*θ_j_*)	0.1	0.4	0.4	0.1	
*P*(*x*|*y_j_*)	0	0.5	0.5	0	log(10/4) = 1.32
*P*(*x*|*y_j_*) *= P*(*x*|*θ_j_*)	0.1	0.4	0.4	0.1	0.2log(10) + 0.8log(10/4) = 1.72

**Table 3 entropy-27-00684-t003:** Neal and Hinton’s mixture model example.

	True Model’s Parameters			Initial Parameters		
	*μ**	*σ**	*P**(*y*)	*μ*	*σ*	*P*(*y*)
*y* _1_	46	2	0.7	30	20	0.5
*y* _2_	50	20	0.3	70	20	0.5

**Table 4 entropy-27-00684-t004:** A mixture model whose *F*″ and *Q* decreased in the convergent process.

	The True Model’s Parameters			Initial Parameters		
	*μ**	*σ**	*P**(*Y*)	*μ*	*σ*	*P*(*Y*)
*y* _1_	40	15	0.5	40	5	0.5
*y* _2_	75	15	0.5	80	5	0.5

## Data Availability

No new data were created or analyzed in this study, date sharing is not applicable.

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
