# Peer review of "Improving the Minimum Free Energy Principle to the Maximum Information Efficiency Principle"

_entropy, 2025, doi:10.3390/e27070684_

Round 1

Reviewer 1 Report (New Reviewer)

Comments and Suggestions for Authors

I thank the author for the interesting paper and the addition it represents to the author's line of work.  The derivation of a novel approach to information and probability merit further work.  I also appreciate the effort to approach active inference and the free energy principle.  The conceptual clarifications on free energy and its interpretation are helpful, and in particular, I find it to be a major contribution to identify the quantity R-G that monotonically decreases during learning, distinct from previous measures like free energy F.  This is explained sufficiently and demonstrated in a simple but illustrative example of maximizing information efficiency.  

While the author makes the point that VFE is inconsistent with physical free energy, my interpretation of Friston's work (e.g. Friston et al 2020 in Entropy, doi 10.3390/e22050516) is that the free energy is intended to characterize the expected work required by an agent in an environment, as part of its homeostatic process.  So, while the author's point may be correct (I did not have time to verify the calculations) I believe that this point merits further development within a agent-environment framework.

The use of semantics via the P-T framework is is appreciated and well illustrated in the examples.  I would however suggest the author consider developing the approach in a case where the agent must learn the semantics with respect to its environment.  In that case, the various constraint functions are not necessarily known.  It might be interesting, for interest, to use a deep neural network to generate the constraints, and then learn progressively to solve the task.  

When comparing to Friston's work on active inference and its implementation, I am also reminded that his work make frequent use of the Laplace approximation for complex free energy functions.  The neurobiological plausibility therein lies in the Hessian resembling a connectivity matrix in the brain.  I wonder if the author here has some comments on basic neurobiological plausibility of the G theory in such terms?

Lastly, thanks to the author for providing reproducible figures. I can confirm that the code provided works correctly (in the interest of time I only looked at the script for Fig6).  My only comment is that it may be useful to invest time in implementing the authors ideas in more popular machine learning frameworks like PyTorch or JAX, so that the ideas become easier to reuse by practitioners.  Personally, this is the first task I would need to do in order to incorporate these ideas into my own modeling work.  Considering the flexibility of the constraints in the P-T framework, this would be quite useful to have as e.g. an easily installable Python library.  But I appreciate that this is not the goal of the author at this point in time.

Comments on the Quality of English Language

I find the manuscript to have some very innovative material regarding the free energy principle and its theoretical background as it relates also to active inference.  Unfortunately, there is an significant amount of technical detail throughout, and sometimes it is challenging to follow the argument in the text.  Without reducing the rigor, I would suggest developing the broad theoretical arguments in to a few coherent paragraphs of clearly argued text, and then follow up with the technical arguments, to make it more accessible for readers.  For instance, when introducing the P-T framework for semantic information theory, there's no mention of biological organism, and only 9 pages later, in section 6.3, does author say "bio-organisms have purposes, ...".  I think this is a great point (i.e. organism homeostasis confers semantics to its information) which is difficult to read because its scattered.   In summary, I think the author has some strong theoretical arguments marred by poor organization.

There are several typographical oddities that don't detract from readability but could be fixed.  For instance, the conventional term for Friston's principle is Free Energy Principle (FEP) not Minimum Free Energe, at least in the literature from Friston and his colleagues.  Another is, e.g. 7.1.3 "A Mixture Model Hard to Converges" would probably be written as "A mixture model with poor covergence" or similar.  Anyway, not a big deal.

Author Response

Please see the pdf file.

Reviewer 2 Report (New Reviewer)

Comments and Suggestions for Authors

Please see the attached manuscript for my comments. 

Author Response

Please see the pdf file.

Round 2

Reviewer 1 Report (New Reviewer)

Comments and Suggestions for Authors

We thank the author for considering the review and revising the manuscript accordingly. 

This manuscript is a resubmission of an earlier submission. The following is a list of the peer review reports and author responses from that submission.

Round 1

Reviewer 1 Report

Comments and Suggestions for Authors

This paper presents some interesting variations of the minimum free energy principle and how inferential algorithms can be adjusted to improve performance.

I suggest to include the work of Silverstein and Pimbley ("Minimum-free energy method of spectral estimation: autocorrelation-sequence approach", Jour. OSA, vol. 7, Is. 3, pp. 356-372, 1990) in the background considerations. 

The paper can be considered as some food for thought. The approach is based on some empirical improvements observed in practical implementations of the resulting algorithms. I believe the presented conclusions tend to be somewhat categorical and poorly justified (cf., the draft's last paragraph). The argumentation should be improved. Nevertheless, the paper brings some interesting considerations for a research area that is current and important to understand the role of entropy and energy in modern data analysis and statistical inference. I recommend that the author revise some of the more categorical statements to highlight the exploratory nature of the work.

The presentation is clear and the use of the language is good.

Reviewer 2 Report

Comments and Suggestions for Authors

please see file attached

Round 2

Reviewer 1 Report

Comments and Suggestions for Authors

The paper has been substantially improved.

In my view, it is ready for publication.

Author Response

Please see the pdf file.

Reviewer 2 Report

Comments and Suggestions for Authors

Author Response

Please see the pdf file.

Round 3

Reviewer 2 Report

Comments and Suggestions for Authors

Dear editor,

I have read the revised version (v3) of the manuscript. 

Whilst a substantial number of changes have been made, some equations removed and some notations clarified, the latter was only a very basic requirement for the paper to be readable at all. In substance however, the author has not addressed at all my criticisms and suggestions. In my opinion, since version 1 the development proposed in the relation to physical entropy does not make sense. The proposed expressions are not justified, even in the third version of the manuscript. For example, Eqs.(27)-(29) still make no sense to me, and I have already explained why. In physics one does not venture to state "let's denote this the prior for that" etc... there are clear equations to start from and the author does not seem to know or want to follow them. In physics, equations pertaining to probabilities associated to the energy in the aforementioned equations should come from equations (24) and (25). 

I have already highlighted for the editor and the author what I thought were the strengths of the paper and what deserved publication (the critique and analysis of FEM), but I find the grander claims being made (on evolution, physics and living organisms) to be neither credible with the proposed analysis, nor fitting at the right place with the main development of the paper -- even though the author tried to enforce this theme more throughout the paper in this 3rd version following my 2nd report.

For the above reasons, I am still not recommending publication of the manuscript. To be a bit more clear about my position. I am recommending rejection of the paper after three iterations essentially dismissing the substance of the my critique which is that the proposed physics analysis and analogy do not hold up to what is usually done in physics and that the proposed claims regarding evolution are not particularly supported (or only very loosely so) by the proposed analysis.

In my opinion, only if the paper sticks to its demonstrable strengths should publication be considered. 

Author Response

Please see the pdf file.
